# Isolation and Genomic Analysis of a Case of *Staphylococcus argenteus* ST2250 Related to Sepsis in Italy

**DOI:** 10.3390/microorganisms12071485

**Published:** 2024-07-20

**Authors:** Giulia Gatti, Francesca Taddei, Anna Marzucco, Maria Sofia Montanari, Giorgio Dirani, Silvia Zannoli, Laura Grumiro, Martina Brandolini, Claudia Colosimo, Laura Dionisi, Ludovica Ingletto, Alessandra Mistral De Pascali, Alessandra Scagliarini, Vittorio Sambri, Monica Cricca

**Affiliations:** 1Department of Medical and Surgical Sciences—DIMEC, Alma Mater Studiorum, University of Bologna, 40138 Bologna, Italy; martina.brandolini3@unibo.it (M.B.); claudia.colosimo2@unibo.it (C.C.); laura.dionisi2@unibo.it (L.D.); ludovica.ingletto@unibo.it (L.I.); alessandra.depascal3@unibo.it (A.M.D.P.); alessand.scagliarini@unibo.it (A.S.); vittorio.sambri@unibo.it (V.S.); monica.cricca3@unibo.it (M.C.); 2Unit of Microbiology, The Great Romagna Hub Laboratory, 47522 Cesena, Italy; francesca.taddei@auslromagna.it (F.T.); anna.marzucco@auslromagna.it (A.M.); sofi.monta.msm@gmail.com (M.S.M.); giorgio.dirani@auslromagna.it (G.D.); silvia.zannoli@auslromagna.it (S.Z.); laura.grumiro@auslromagna.it (L.G.)

**Keywords:** *Staphylococcus argenteus*, NGS, bloodstream infections, *Staphylococcus aureus*, genomic analysis

## Abstract

*Staphylococcus argenteus*, identified in 2006, represents a challenging case of bacterial taxonomic identification because of its high similarity to *Staphylococcus aureus*. In this context, neither mass spectrometry (MS) nor 16S gene analysis cannot precisely reveal the difference between the two species. In our study, the sensitivity to antibiotics of *S. argenteus* isolated from blood culture was tested, and the investigation of the bacterial genome was performed by Multi-Locus Sequence Typing (MLST) and Whole-Genome Next-Generation Sequencing (WG-NGS). The pathogen was identified as ST2250 and presented perfectly matched resistance genes, namely *aph(3′)-III*, *mgrA*, and *sepA*, whereas the virulence gene detected was *scn*. Two plasmids were found: the pSAS plasmid, belonging to the family of Inc18, and plasmid pN315, belonging to the Rep3 group. The epidemiological distribution and the spread of *S. argenteus* infection are scarcely documented, particularly when associated with sepsis. Therefore, a correct taxonomy identification, antibiogram, and resistance gene analysis may help in acquiring knowledge about this bacterium and implement its detection and treatment.

## 1. Introduction

In 2006, a new presumable lineage of *Staphylococcus aureus*, assigned to Clonal Complex 75 (CC75), was isolated from samples from individuals belonging to indigenous communities in the Northern Territory, Australia. Multi-Locus Sequence Typing (MLST) proved this was a very interesting and challenging case study in bacterial systematics, so much so that CC75 was renamed *Staphylococcus argenteus* [1,2]. The phenotypic difference between *S. aureus* and *S. argenteus* is a non-pigmented appearance of colonies that is due to the lack of the crtOPQMN operon, essential for the synthesis of staphyloxanthin, a carotenoid pigment [3]. Through genomic investigation, *S. argenteus* can be considered an ancestor of *S. aureus,* with which it shares about 95% of its Average Nucleotide Identity (ANI), although 16S gene results are not distinguishable between the two. A difference can be detected by studying the *nuc*, *gap*, *rpoB*, *sodA*, *tuf,* and *hsp60* genes or performing MLST [4,5]. Worldwide, *S. argenteus* is concentrated in three “hotspot” regions, namely Southeast Asia, Australia, and the Amazon, but has also been reported in New Zealand, Fiji, America, and, in rare cases, in Europe [5,6]. In subtropical areas, *S. argenteus* is considered an emerging pathogen among patients with community-onset staphylococcal infections; linked to this phenomenon, *S. argenteus* seems to have recently also been found in healthcare-associated infections among recently hospitalized patients [7]. Besides its spread, the clinical implication of *S. argenteus* relates to the presence of genes encoding staphylococcal enterotoxins, hemolysin, and Panton–Valentine leucocidin [8]. Additionally, public health systems have been warned about the antimicrobial resistance genes reported in *S. argenteus* isolates, since penicillin (*blaZ*-positive) and methicillin (mostly *mecA*-positive)-resistant bacteria similar to methicillin-resistant *S. aureus* (MRSA) have been described in detail [9,10]. Although most *S. argenteus* cases have been found to be sensitive to methicillin, some isolates among patients in Aboriginal communities in Northern Australia affected by impetiginous lesions have been described as resistant to this antimicrobial agent [7,8]. In view of healthcare-associated infections, the acquisition of resistance to methicillin by *S. argenteus* draws attention to high antibiotic pressure and its further implications in public health [7]. *S. argenteus* is known to be a sporadic pathogen causing skin and soft tissue infections, bone infections, and sepsis [11,12], cases in which it can be misdiagnosed as *S. aureus* bacteremia. In light of this, the clinical conditions of patients with *S. argenteus* bacteremia can worsen if they develop lower respiratory tract infections, which increase mortality compared to methicillin-sensitive *Staphylococcus aureus* (MSSA) infections [7,13]. A plausible explanation for this enhanced trend relates to the acquisition of species-specific *S. argenteus* virulence factors [7]. In this study, we describe the first *S. argenteus* case isolated at the Greater Romagna Hub Laboratory from a blood culture, which, in our knowledge, represents the first case of *S. argenteus* bacteremia described in Italy.

## 2. Materials and Methods

### 2.1. Sampling

The blood sample was collected in an emergency room at Riccione and delivered to the Greater Romagna Hub Laboratory in BACT/ALERT^®^ culture media (bioMérieux, Marcy l’Etoile, France). The sample was incubated for about 22 h until positivity was declared. The sample was processed according to the diagnostic routine of the laboratory.

### 2.2. Identification and Antibiogram

Then, the blood culture was seeded on Columbia + 5% sheep blood, COS (bioMérieux, Marcy l’Etoile, France). The identification of the isolated colonies was performed using Vitek^®^ MS (bioMérieux, Marcy l’Etoile, France) and resulted in *S. argenteus* identification; since it was the first case detected in our laboratory, the isolate was tested for antimicrobial sensitivity and with an NGS approach. The Antibiotic Sensitivity Test (AST) was performed according to EUCAST’s breakpoint guidelines [14] calculated using MicroScan WalkAway (Beckman Coulter Inc., Brea, CA, USA). The isolated colony presented a peculiar feature: it resembled a *S. aureus* colony, but without pigmentation. For this reason, the colony was selected for an in-depth investigation.

### 2.3. Sequencing

To confirm the mass spectrometry spectrum, the whole bacterial genome was investigated. The sample was extracted using the automated system of Maxwell^®^ CSC Pathogen Total Nucleic Acid Kit (Promega Corporation, Madison, WI, USA). The library was prepared using the DNA Prep kit (Illumina Inc., San Diego, CA, USA) according to the manufacturer’s instructions and sequenced on MiSeq platform (Illumina Inc., San Diego, CA, USA). The raw sequence FastQ files were deposited in the Sequence Reads Archive (SRA) (BioProject ID: PRJNA1032758).

### 2.4. Bioinformatical Analysis and Confirmation

Forward and reverse reads were controlled using FastQC v.0.12.1 (Babraham Institute, Cambridge, UK), trimmed using Trimmomatic v.0.38.1 by using default parameters, and de novo assembled using SPAdes v.3.13.0 with default parameters. The confirmation of pathogen identification was performed by using a ribosomal MLST (rMLST) on 53 target genes, and then the MLST profile was traced using Ridom SeqSphere+ v.9.0 with 7 target genes described in the literature (*arcC*, *aroE*, *glpF*, *gmk*, *pta*, *tpi,* and *yqiL*). The presence of genes involved in resistance mechanisms was investigated using the ResFinder v.4.1 and Comprehensive Antibiotic Resistance Database (CARD v.3.2.8) databases. Additionally, the PlasmidFinder v.2.0 database was queried to search for the presence of plasmids, and the Virulence Factor Gene Database to screen for additional virulence factors. The mobile genetic elements (MGEs) were investigated by the MobileElementFinder database v.1.0.3.

Once the profile of the resistance genes was obtained, phenotypic resistance was confirmed for the corresponding antibiotic by performing ETEST^®^ (bioMérieux, Marcy l’Etoile, France) and MicroScan WalkAway in triplicate. To complete the phenotypic resistance pattern, sensitivity to fosfomycin was evaluated by the agar dilution method AD Fosfomycin 0.25–256 assay (Liofilchem srl, Roseto degli Abruzzi, Italy). Resistance was declared according to EUCAST’s guidelines [14]. To better understand the evolution similarity of the isolate, 15 complete *S. argenteus* genomes were downloaded from the NCBI database and a phylogenetic tree was inferred in MEGA11 [15] using a UPGMA method and the evolutionary distances were computed using the Maximum Composite Likelihood method. The tree was visualized using FigTree v.1.4.4 (accessed on 10 July 2024: http://tree.bio.ed.ac.uk/software/figtree/) and was midpoint-rooted.

### 2.5. S. argenteus MSHR1132

The *S. argenteus* genome (NCBI RefSeq Assembly: GCF_000236925.1) was downloaded from NCBI and included in the analysis. The sequence belonged to the MSHR1132 strain and was deposited in November 2011 by The Wellcome Trust Sanger Institute. There are 2689 annotated genes, including 2533 protein-coding genes. The bacterium was isolated from a blood culture of a woman affected by necrotizing fasciitis and the genome was classified in an ST 1850 [2].

## 3. Results

### 3.1. rMLST and MLST

The rMLST resulted in rST 4649, typically associated with the *S. argenteus* genomic profile. According to the MLST, the genome sequence is classified in an ST 2250 and target genes attributed to its alleles (Table 1). According to the genomes deposited in the public database PubMLST (https://pubmlst.org/, accessed on 16 July 2024), 111 isolates were recorded with rST 4649 and 3 with ST 2250.

### 3.2. Antibiogram

The antibiogram performed in microdilution showed bacterial sensitivity to amikacin (≤4 mg/mL), ceftaroline (0.25 mg/mL), ciprofloxacin (≤0.25 mg/mL), clindamycin (≤0.12 mg/mL), daptomycin (≤0.25 mg/mL), erythromycin (≤0.5 mg/mL), gentamycin (≤1 mg/mL), levofloxacin (≤0.5 mg/mL), linezolid (≤2 mg/mL), minocycline (≤1 mg/mL), moxifloxacin (≤0.25 mg/mL), oxacillin (≤0.25 mg/mL), penicillin (≤0.03 mg/mL), teicoplanin (≤1 mg/mL), tetracycline (≤1 mg/mL), tigecycline (≤0.25 mg/mL), tobramycin (≤1 mg/mL), trimethoprim and sulfamethoxazole (≤1/19 mg/mL), and vancomycin (1 mg/mL). Therefore, the isolate did not present any resistance to the tested antibiotics. In addition, the agar dilution method for the detection of fosfomycin resistance resulted in an MIC of 8 mg/mL; therefore, the isolate was sensitive to the antibiotic according to EUCAST’s guidelines (Table 2).

### 3.3. Resistance Genes, Plasmids, and Virulence Genes

At a genomic level, the resistance gene identified by both the queried databases (ResFinder and CARD) had a perfect match in terms of identity and sequence length with *aph(3′)-III* (aph(3′)-III_M26832), which can inactivate aminoglycoside antibiotics such as amikacin, butirosin, kanamycin, neomycin, lividomycin, paromomycin, and ribostamycin. The CARD database found two other additional perfectly matched genes: *mgrA* (inducing resistance to fluoroquinolones, cephalosporins, penams, tetracyclines, peptide antibiotics, and disinfecting and antiseptic agents) and *sepA* (inducing resistance to disinfecting and antiseptic agents). Other sequences that are probably responsible for resistance mechanisms were detected, even though there was no perfect match: *blaZ* (ResFinder and CARD) (96.8% identity and 100% length), *arlR* (98.63% identity and 100% length), *norC* (97.4% identity and 100% length), *Staphylococcus aureus fosB* (97.12 identity and 100% length), *Staphylococcus aureus lmrS* (90.44% identity and 100% length), *mepR* (92.81% identity and 100% length), *sdrM* (89.04% identity and 100% length), and the *vanT* gene in the *vanG* cluster (34.23% identity and 55.48% length) (CARD). Given the presence of *blaZ*, resistance to penicillin was confirmed, with an MIC of 0.125 mg/mL. Indeed, according to EUCAST’s guidelines, the isolate presented resistance to penicillin.

Also, to better understand the isolate’s resistance and invasiveness capability, the following plasmids were investigated: the pSAS plasmid (reference sequence BX571858; 20,652 base pairs, bp) with the *rep16* gene and belonging to the family of Inc18 was found, with an identity and length match of 100% for Coding Sequence 8. In addition, the plasmid pN315 (reference sequence AP003139; 24,653 bp) with the *rep5a* gene and belonging to the plasmid family Rep3 was detected, with a length match of 100% and an identity percentage of 99.88% (one single-nucleotide polymorphism) in the Coding Sequence of SAP001. Regarding the virulence genes, a perfect match in length and sequence was declared with the *scn* gene, a staphylococcal complement inhibitor, and a 99.6% identity match was found with the *sak* gene encoding staphylokinase. Just one MGE was found: ISSau5 (NC_002952.2), belonging to the IS30 family, an insertion sequence of 1136 nucleotides. The evolutionary relation of the 16 genomes was calculated and the original tree, with a total of 2,834,926 positions, was analyzed. The isolate’s genome is most likely related to the *S. argenteus* MSHR1132 strain (Figure 1).

Similarity with the *S. aureus* reference genome (GCF_000013425.1) was calculated using Average Nucleotide Identity and was 87% (Figure 2).

## 4. Discussion

Since its first isolation, *S. argenteus* has been documented as a probable blood stream contaminant; in particular, ST2250 was found to cause sepsis [16,17]. In the Great Romagna Hub Laboratory, a positive blood culture isolate was classified as *S. argenteus* with the use of mass spectrometry; this was then confirmed with ANI analysis and MLST. The presence of resistance genes was investigated through the ResFinder and CARD databases: the two were concordant on the presence of *aph(3′)-III* (aph(3′)-III_M26832), a gene encoding aminoglycoside-3′′-O-phosphoryltransferase III and responsible for the inactivation of aminoglycoside antibiotics by catalyzing acetyltransferase and phosphotransferase reactions [18]. *mgrA*, only found in CARD, controls over 300 genes involved in antibiotic resistance and virulence, including an upregulation of exotoxins and capsular polysaccharide [19]. Moreover, *sepA* has been described in staphylococci encoding efflux pumps related to biocide resistance [20]. The other detected sequences did not present a perfect match with the genes deposited in the databases, even though their presence can be investigated in more depth. Particularly, the knowledge of the role of *blaZ,* found in the genome of *S. argenteus,* in regulating penicillin resistance can be extended to the identification of the gene’s repressors (*blaI*) and regulators (*blaR1*) [21]. Particularly, genomic analysis can reveal a resistance profile which may be missed or misdiagnosed due to technical errors during the laboratory workflow. This suggests that a collaboration between molecular biology techniques and standard wet-lab methodology can trace the pathogen resistance trend more precisely. In addition, the presence of the *vanT* gene, grouped in the *vanG* cluster, can be related to a peculiar pattern of vancomycin resistance [22,23]. Indeed, the antibiogram result obtained for vancomycin was considered dubious, and, according to EUCAST’s guidelines [14], which suggest a breakpoint of 2 mg/L, the MIC that was found was reduced, at 1 mg/L. Other investigations are needed to confirm the presence of the *vanT* sequence and to understand its impact on MIC. A crucial role in the acquisition of antimicrobial resistance is played by plasmids; indeed, ST2250 has been proven to possess the plasmid p2_1801221, with *rep5a* and *rep16* involved as determinants in penicillin resistance due to carrying the *blaZ* gene [9]. In *S. argenteus*, the *blaZ* gene is more likely linked to plasmid acquisition than to chromosomal origin. The study of Aung et al., according to Olsen’s classification, divided plasmid-related *blaZ* in *S. argenteus* into P1 and P2 subgroups; this subdivision may describe a multiple origin of *blaZ* [24]. In particular, the Inc18 plasmid family is commonly isolated from bacteria that cause nosocomial infections and encodes resistance to different antimicrobial agents such as macrolides, lincosamides, streptogramins, and vancomycin. Among staphylococci, Inc18 plasmids carrying the *vanA* gene are responsible for vancomycin resistance acquired from enterococci. This acquisition can likely be correlated with poor hygiene conditions, and in this setting, the rare vancomycin-resistant *S. aureus* is generally also resistant to methicillin and derives its *vanA*-mediated resistance from enterococci. Nevertheless, vancomycin resistance incidence in MRSA was revealed to be a serious threat not only for the treatment of *S. aureus* infections but also for all *S. aureus* complex-related bacterial species [25]. In the same manner, the *scn* virulence gene, due to its role as an immune evasion marker, may be linked to animal-to-human transmission and is generally found in MSSA [26]. The presence of the *scn* gene raises the need to take strict hygiene controls in animal-related and animal-derived products [27]. According to the One Health epidemiological approach, the investigation and the understanding of the resistance and virulence genes acquired by *S. argenteus* may be relevant not only to the analysis of their distribution but also to the prevention and treatment of the human healthcare-associated matter in question. More precisely, bloodstream infections may be fatal for patients and indeed a precise identification of the infective agent, its virulence, and resistance are necessary both for the monitoring of the spread and for the control of the patient’s infection. The single MGE found in the sequence was previously described in MRSA252, a hospital-acquired pathogen [28]. Considering the increase in sepsis incidence in recent years and the variety of its causative agents, a new frontier of research opens in correlating the infection with gut microbiota dysbiosis [29]. The alteration of gastrointestinal mucosa integrity allows for the infiltration of various entities, including bacteria [30]. Together with other commensal microorganisms, *S. aureus* may be found to be an intestinal colonizer pathogen that enters the bloodstream and causes diffuse infections [31]. Due to its large similarity with *S. aureus*, it can be assumed that *S. argenteus* could present the same outcome, but other studies are needed to confirm its presence in the gut microbiota. On the other hand, new evidence has been reported in the literature about *S. argenteus* isolation from blood cultures and its severity for patients [12]. Due to the scarcity of information, its antibiotic resistance and geographical distribution need to be traced and monitored. In Australia, in 2023, *S. argenteus* ST 2250 was isolated in a patient without prior exposure and was found to be resistant to daptomycin [32]. Similarly, in Korea, *S. argenteus* ST 8342 and 8343 lead to severe clinical outcomes in both community- and healthcare-acquired infections [33]. Therefore, studying the epidemiological trend of *S. argenteus* by using molecular assays or NGS helps not only in characterizing the pathogen but also in preventing patient complications [13].

## 5. Conclusions

The assortment of human pathogens requires continuous monitoring of novel deleterious mutations and causative agents to prevent severe sequelae. Investigating and understanding the phenotypic and genotypic characteristics of uncommon infections is significant in order to organize an efficient diagnostic workflow and administer the proper treatment. In this scenario, as far as the authors know, this is the first documented case of *S. argenteus* causing sepsis in Italy.

## Figures and Tables

**Figure 1 microorganisms-12-01485-f001:**
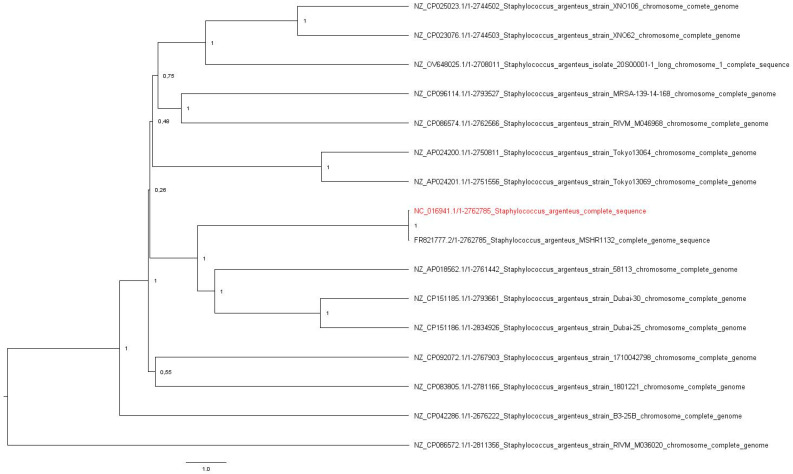
The original tree resulting from the UPGMA analysis. The evolutionary distances among 16 genomes were calculated with the Composite Likelihood Method using Mega 11 v.11.0.13. The tree was midpoint rooted using FigTree v.1.4.4. The isolate’s genome is highlighted in red.

**Figure 2 microorganisms-12-01485-f002:**
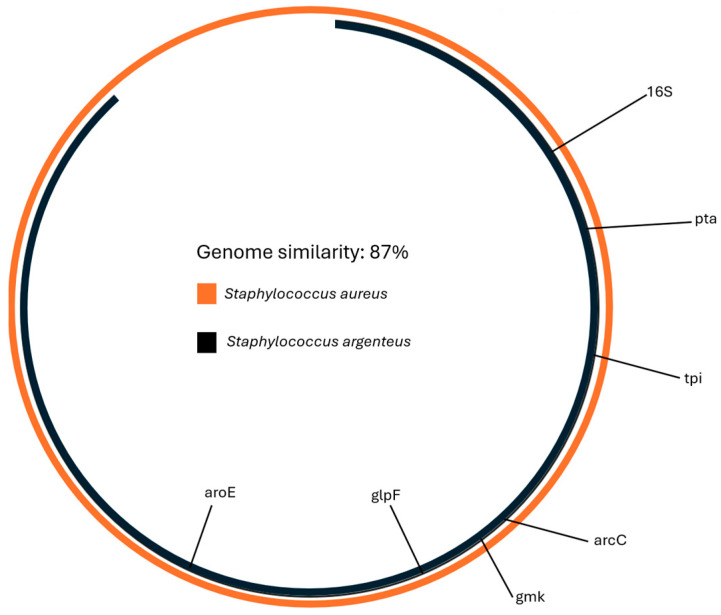
The similarity between S. aureus and the genome of the isolate was 87%. The similarity was calculated with Average Nucleotide Identity analysis.

**Table 1 microorganisms-12-01485-t001:** Results of MLST of Staphylococcus argenteus isolated from blood culture at the Greater Romagna Hub Laboratory. The 7 target genes were classified, and the sequence type was determined.

Genes	Allele
*arcC*	151
*aroE*	325
*glpF*	215
*gmk*	34
*pta*	175
*tpi*	180
*yqiL*	169
ST *	2250

* ST = sequence type.

**Table 2 microorganisms-12-01485-t002:** The antibiogram results of Staphylococcus argenteus. The isolate was sensitive to 20 antibiotics tested.

Antibiotic	MIC * (mg/mL)	Result
Amikacin	≤4	Sensitive
Ceftaroline	0.25	Sensitive
Ciprofloxacin	≤0.25	Sensitive
Clindamycin	≤0.12	Sensitive
Daptomycin	≤0.25	Sensitive
Erythromycin	≤0.5	Sensitive
Gentamycin	≤1	Sensitive
Levofloxacin	≤0.5	Sensitive
Linezolid	≤2	Sensitive
Minocycline	≤1	Sensitive
Moxifloxacin	≤0.25	Sensitive
Oxacillin	≤0.25	Sensitive
Penicillin	≤0.03	Sensitive
Teicoplanin	≤1	Sensitive
Tetracycline	≤1	Sensitive
Tigecycline	≤0.25	Sensitive
Tobramycin	≤1	Sensitive
Trimethoprim/Sulfamethoxazole	≤1/19	Sensitive
Vancomycin	1	Sensitive
Fosfomycin	8	Sensitive

* Minimum Inhibitory Concentration.

## Data Availability

The original data presented in the study are openly available in BioProject. Accession Number: PRJNA1032758.

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
