# Peer review of "Isolation and Genomic Analysis of a Case of Staphylococcus argenteus ST2250 Related to Sepsis in Italy"

_microorganisms, 2024, doi:10.3390/microorganisms12071485_

Round 1

Reviewer 1 Report

Comments and Suggestions for Authors

The submitted manuscript focuses on Staphylococcus argenteus, gram-positive cocci from the Staphylococcus genus that were identified from a 55-year-old Indigenous Australian female's blood culture in Darwin, Northern Territory, Australia, in 2006. The species is closely related to S. aureus, hence separation is difficult. Staphylococcus argenteus is harmful to human cells due to its high production of alpha-hemolysin. To the best of Authors (and mine) knowledge, this is the first such case reported in Italy. Therefore, the topic of this work is valid and important.

The submitted manuscript is a case report (communication), however according to the information stated in the Special Issue description ”short communications are invited” so it is OK.

Due to the “communication” character of this manuscript, my review can’t be too long as there is simply not much text to be reviewed. However, I think that the work can benefit from considering the comments below.

Major comments:

Line 29, number of keywords must be increased to at least 5-6, I recommend “Staphylococcus aureus”, “genomic analysis”

Line 56, I guess, it is similar to MRSA? This should be stated.

Lines 69-75, some more information about the patient should be included

Lines 117-123, those kind of results are usually presented in a form of a Figure/Table. Please consider this option.

Lines 208-209, I understand that this is just a communication, yet a short “conclusion” paragraph would be appreciated. It is for the readers benefit.

Minor issues:

Line 109, it should be “S. argenteus

Lines 210-211, this part must be removed

Author Response

The authors thank the Reviewer 1 for their interesting revisions and useful comments and hope to have completed the assignment to elevate the level of this manuscript. Here, the authors report the responses to all the suggestions: 

Major comments:

Comment: Line 29, number of keywords must be increased to at least 5-6, I recommend “Staphylococcus aureus”, “genomic analysis”

Response: The recommended keywords were added.

Comment: Line 56, I guess, it is similar to MRSA? This should be stated.

Response: The sentence was stated.

Comment: Lines 69-75, some more information about the patient should be included

Response: Since S.argenteus is an emergent pathogen not fully understood, the aim of the study was to characterize the isolate at phenotypic and genomic level. The authors focused on the analysis of the bacterium to warn to the scientific community that the strain is present in Italy where it can infect patients. Therefore, the authors traced a profile to understand the risk associated to S.argenteus characteristics in terms of resistance and virulence. On the other hand, if Reviewer 1 believes that additional clinical information is needed, the authors should ask for more days to individuate and contact the patient, since the analysed isolated colony was anonymized.

Comment: Lines 117-123, those kind of results are usually presented in a form of a Figure/Table. Please consider this option.

Response: the table was added as Table 2 (line 140).

Comment: Lines 208-209, I understand that this is just a communication, yet a short “conclusion” paragraph would be appreciated. It is for the readers benefit.

Response: The "Conclusion" paragraph was added (line 249).

Minor issues:

Comment: Line 109, it should be “S. argenteus

Response: The term was corrected.

Comment: Lines 210-211, this part must be removed

Response: the part was removed.

Reviewer 2 Report

Comments and Suggestions for Authors

Thank you for submitting the manuscript "Isolation and genomic analysis of a case of Staphylococcus argenteus ST2250 related to sepsis in Italy" to Microorganisms. The manuscript is interesting, but it reports few experimental results. The results that were reported are clearly exposed and duly discussed.

Author Response

Comment: Thank you for submitting the manuscript "Isolation and genomic analysis of a case of Staphylococcus argenteus ST2250 related to sepsis in Italy" to Microorganisms. The manuscript is interesting, but it reports few experimental results. The results that were reported are clearly exposed and duly discussed.

Response: The authors thank the Reviewer 2 for their time spent to review the manuscript. The authors thank also Reviewer 2 to have considered the study interesting and suitable for a publication.

Reviewer 3 Report

Comments and Suggestions for Authors

The manuscript submitted by Gatti et al., titled: "Isolation and genomic analysis of a case of Staphylococcus argenteus ST2250 related to sepsis in Italy", is an interesting genomic analysis study that focuses on Staphylococcus argentheus ST2250 related to sepsis in the setting of Italy. The manuscript is well written and organized dealing with an interesting topic of distinction between S. argenteus and S. aureus. This has significant potential clinical implications for designing optimal treatment.

The reviewer would like to raise the following points for the authors' consideration:

1. It would be helpful to include some information on the original study where the samples were derived from as leftovers. Provide characteristics of participants, inclusion/exclusion criteria and rationale of sampling and sample size.

2. It would be helpful to provide a graph indicating the level of similarity (as a map or diagram) between the S. argenteus and S. aureus

3. Consider discussing how/if the gut microbiome may influence clinical outcomes (sepsis)

4. Consider discussing in greater depth the clinical significance and application potential.

5. The template of the manuscript needs to be cleaned up.

6. While the topic is relevant understudied it would help the manuscript if there were more references supporting it.

Comments on the Quality of English Language

The English language would benefit from the proofreading by a native English speaker.

Author Response

The authors thank the Reviewer 3 for their interesting revisions and useful comments and hope to have completed the assignment to elevate the level of this manuscript. Here, the authors report the responses to all the suggestions: 

Comment: It would be helpful to include some information on the original study where the samples were derived from as leftovers. Provide characteristics of participants, inclusion/exclusion criteria and rationale of sampling and sample size.

Response: The sample was not included in a study, but it was found during the laboratory daily diagnostic routine at the Operative Unit of Microbiology. Due to the peculiar features of the seeded colony, the authors decided to investigate in depth the pathogen and this manuscript reports the results of the genomic analysis. The authors clarified the collection of the specimen (line 71-86)

Comment: It would be helpful to provide a graph indicating the level of similarity (as a map or diagram) between the S. argenteus and S. aureus

Response: The figura was added as Figure 2 (line 178).

Comment: Consider discussing how/if the gut microbiome may influence clinical outcomes (sepsis)

Response: The authors discussed the influence of gut microbiota (line 235-242)

Comment: Consider discussing in greater depth the clinical significance and application potential.

Response: The authors discussed the clinical significance more in depth (line 242-251).

Comment: The template of the manuscript needs to be cleaned up.

Response: The template was cleaned up by adding subheadings to facilitate readers and checking for typos.

Comment: While the topic is relevant understudied it would help the manuscript if there were more references supporting it.

Response: The authors added 7 references, 4 regarding specifically for S. argenteus infection.

Reviewer 4 Report

Comments and Suggestions for Authors

The authors present an interesting study examining a strain of staphylococcus argentus; a strain of bacteria that presents with a striking similarity to staphylococcus aureus, making it a challenge to accurately identify and indeed track its pathogenic spread within populations. Following identification at a clinical setting, the authors profile the strain to determine unique features of such which may enable improved tracking whilst also adding to the knowledge on the biology of the unique strain.

In reviewing the manuscript, I made a couple of observations. The following should be considered by the authors when preparing a suitable revision.  

1.      The writing of the manuscript is for the most part of a high standard, however, there are some instances where there are small typos in terms of the language used. For example, in Line 60 the phrase ‘in the light of this’ is more commonly delivered as ‘in light of this’. This is just one of a couple of instances where the manuscript can be tidied up in terms of language and grammar, and the authors are encouraged to do another review of the writing to ensure these instances are as a few as possible.

2.      The methods section needs to be broken down into subheadings. In its current form the information is too dense and needs to be subsectioned with respect to each approach to the analyses.

3.      In the same vein as the previous point, the results method would benefit from having subsections to break the individual results apart.

4.      The formatting of figure 1 should be revised. The dimensions and scaling make the information too small to interpret cleanly.  

Comments on the Quality of English Language

As mentioned in my report, there are some grammatical and language aspects, while relatively minor in nature, could be improved upon to bring this to publication standard. 

Author Response

The authors thank the Reviewer 4 for their interesting revisions and useful comments and hope to have completed the assignment to elevate the level of this manuscript. Here, the authors report the responses to all the suggestions:

Comment: The writing of the manuscript is for the most part of a high standard, however, there are some instances where there are small typos in terms of the language used. For example, in Line 60 the phrase ‘in the light of this’ is more commonly delivered as ‘in light of this’. This is just one of a couple of instances where the manuscript can be tidied up in terms of language and grammar, and the authors are encouraged to do another review of the writing to ensure these instances are as a few as possible.

Response: The authors checked the manuscript for typos and grammar error according to your suggestion.

Comment: The methods section needs to be broken down into subheadings. In its current form the information is too dense and needs to be subsectioned with respect to each approach to the analyses.

Response: the authors subsectioned the Materials and Method section.

Comment: In the same vein as the previous point, the results method would benefit from having subsections to break the individual results apart.

Response:  the authors subsectioned the Results section.

Comment: The formatting of figure 1 should be revised. The dimensions and scaling make the information too small to interpret cleanly.  

Response: the authors changed the dimension of the information in figure 1.

Reviewer 5 Report

Comments and Suggestions for Authors

The manuscript by Gatti et al. entitled "Isolation and genomic analysis of a case of Staphylococcus argenteus ST2250 related to sepsis in Italy" presents the isolation and analysis of a new pathogen, S. argenteus, in Italy using whole-genome sequencing (WGS) with the Illumina technique. The manuscript is interesting, but not without its shortcomings. Due to its small size (results only span two paragraphs or one page), I would recommend categorizing it as a Short Communication.

First and foremost, it would be beneficial to learn more about the clinic and the patient, as the manuscript does not touch upon this topic at all. 

Below are several other issues and suggestions:

Title: The Latin name of the pathogen should be in italics.

Affiliations: Affiliation 3 is not specified. Affiliation 2 lists six authors and seven emails. Is one email redundant?

Lines 78 and 89: Remove the word "Coulter" from the device name.

Line 86: Version for FastQC?

Line 90: Ridom SeqSphere+ should not be italicized or highlighted in color.

Line 90: It is stated as 8 genes, but only 7 are provided.

Line 104: Remove "Molecular Evolutionary Genetics Analysis version 11."

Line 105: The reference should be a number in square brackets.

Line 109: S. argenteus in italics.

Lines 129-155: Italicize the names of all genes and Latin names of microorganisms.

Line 155: In the Materials and Methods section, it is recommended to include information about S. argenteus MSHR1132, its origin, sequence type, and any differences from the strain investigated.

Figure 1: It states MEGA X, but in the Materials and Methods, it mentions MEGA 11.

Figure 1: Italicize the names of the pathogens.

Figure 1: Where is the outgroup?

Line 210: Remove "For research articles with several authors, a short paragraph specifying their individual contributions must be provided. The following statements should be used."

Line 223: Typo: BioProject.

Comments on the Quality of English Language

The English text, however, contains some rough edges, but overall not bad.

Author Response

The authors thank the Reviewer 5 for their interesting revisions and useful comments and hope to have completed the assignment to elevate the level of this manuscript. Here, the authors report the responses to all the suggestions: 

Comment: The manuscript by Gatti et al. entitled "Isolation and genomic analysis of a case of Staphylococcus argenteus ST2250 related to sepsis in Italy" presents the isolation and analysis of a new pathogen, S. argenteus, in Italy using whole-genome sequencing (WGS) with the Illumina technique. The manuscript is interesting, but not without its shortcomings. Due to its small size (results only span two paragraphs or one page), I would recommend categorizing it as a Short Communication.

Response: The authors categorized the manuscript as a Short Communication. 

Comment: First and foremost, it would be beneficial to learn more about the clinic and the patient, as the manuscript does not touch upon this topic at all. 

Response: Since S.argenteus is an emergent pathogen not fully understood, the aim of the study was to characterize the isolate at phenotypic and genomic level. The authors focused on the analysis of the bacterium to warn to the scientific community that the strain is present in Italy where it can infect patients. Therefore, the authors traced a profile to understand the risk associated to S.argenteus characteristics in terms of resistance and virulence. On the other hand, if Reviewer 5 believes that additional clinical information is needed, the authors should ask for more days to individuate and contact the patient, since the analysed isolated colony was anonymized.

Comment: Title: The Latin name of the pathogen should be in italics.

Response: The name of the pathogen was changed in italics. 

Comment: Affiliations: Affiliation 3 is not specified. Affiliation 2 lists six authors and seven emails. Is one email redundant?

Response: The authors corrected the affiliations and the emails.

Comment: Lines 78 and 89: Remove the word "Coulter" from the device name.

Response: The authors eliminated "Coulter" from lines 78 and 89.

Comment: Line 86: Version for FastQC?

Response: The authors added the FastQC versione (Line 98)

Comment: Line 90: Ridom SeqSphere+ should not be italicized or highlighted in color.

Response: The authors corrected the font (Line 102)

Comment: Line 90: It is stated as 8 genes, but only 7 are provided.

Response: The authors corrected the stated gene from 8 to 7. (Line 103)

Comment: Line 104: Remove "Molecular Evolutionary Genetics Analysis version 11."

Response: The authors eliminated "Molecular Evolutionary Genetics Analysis version 11". (Line 117)

Comment: Line 105: The reference should be a number in square brackets.

Response: The authors added the reference in square brackets. (Line 118) (Ref 15)

Comment: Line 109: S. argenteus in italics.

Response: The authors changed the term in italics.

Comment: Lines 129-155: Italicize the names of all genes and Latin names of microorganisms.

Response: The terms were changed in italics.

Comment: Line 155: In the Materials and Methods section, it is recommended to include information about S. argenteus MSHR1132, its origin, sequence type, and any differences from the strain investigated.

Response: The authors added a paragraph (2.5 S. argenteus MSHR1132) where the characteristics of the sequence are described. (Line 123-129)

Comment: Figure 1: It states MEGA X, but in the Materials and Methods, it mentions MEGA 11.

Response: The name of the software was corrected in MEGA 11.

Comment: Figure 1: Italicize the names of the pathogens.

Response: The authors used FigTree to visualize the tree and, unfortunately,the software does not allow to modify the font of the sequence name. It is a limit of the tool implied in the study.  

Comment: Figure 1: Where is the outgroup?

Response: The authors thank the Reviewer 5 for the comment. The tree was midpoint rooted. 

Comment: Line 210: Remove "For research articles with several authors, a short paragraph specifying their individual contributions must be provided. The following statements should be used."

Response: The authors removed the sentence. (Line 272-273)

Comment: Line 223: Typo: BioProject.

Response: The authors corrected the term. (Line 285)